# Evaluating Cultural Impact on Subject-Independent EEG-Based Emotion Recognition Across French, German, and Chinese Datasets

Anshul Sheoran
*Dept. of Applied Computer Science*
*University of Winnipeg*
Winnipeg, MB, Canada
nolastname-a2373@webmail.uwinnipeg.ca

Camilo E. Valderrama
*Dept. of Applied Computer Science*
*University of Winnipeg*
Winnipeg, MB, Canada
c.valderrama@uwinnipeg.ca

*Abstract*—Culture influences emotional expression and recognition, affecting how individuals perceive and regulate emotions. Given this effect of cultural background, previous studies have suggested that incorporating demographic information can enhance emotion recognition in Electroencephalography (EEG) based approaches. However, until now, most studies have focused on improving prediction accuracy, ignoring the extent to which cultural factors impact EEG-based emotion recognition. To address that gap, this study investigates how cultural factors impact emotion prediction by using a stacking model that combines attention mechanism layers with multinomial logistic regression. The attention mechanism layer focused on detecting the cortical areas in which the model paid more attention to predicting the emotions, while the logistic regression analyzed how the cultural factors affect the odds of accurately predicting emotions. To test our model, we used EEG data capturing three emotions (negative, neutral, and positive) from 31 subjects of three nationalities: 15 Chinese, 8 French, and 8 German. Our approach achieved accuracies of 77.3%, 73%, and 65% for recognizing the emotions in the Chinese, French, and German subjects, respectively. Our approach revealed that incorporating cultural information increases the odds of predicting positive emotions for Chinese subjects and negative emotions for French and German subjects. Moreover, French and German subjects exhibited similar neural patterns across emotions, indicating a closer cultural alignment between these groups. Our findings highlight the critical role of cultural context in emotion recognition models. This inclusion not only improves emotion prediction accuracy for subject-independent approaches but also promotes inclusivity and ethical practices in emotion recognition systems.

*Index Terms*—Emotion Recognition, Deep Neural Networks, Attention Mechanism, Cultural influence

## I. Introduction

Emotions are pivotal in human life, significantly shaping our daily experiences and decision-making processes. They add depth and meaning to our relationships. Different environments and cultures shape human physical traits and ways of thinking [1]. This worldwide diversity results in variations and similarities in how individuals experience and express emotions in different cultures [2]. Individuals in cultures with strong social norms may regulate negative emotions to maintain social harmony, while in others, emotions may be more openly displayed [3]. For instance, a study [4] conducted in the United States found that students recognized Caucasian-American emotions more accurately and perceived Japanese expressions as more intense than Caucasian-American expressions, suggesting a cultural bias in emotion recognition.

In recent years, electroencephalography (EEG) has been used to study how emotions and brain activity relate. Unlike facial expressions or speech, EEG provides neural data, offering deeper insights into neuronal emotional responses [5]. Two approaches can be used to build EEG-based emotion recognition systems [6]. One approach is the subject-dependent method, where a model is trained and tested using EEG data from the same individual. The other is the subject-independent method, in which the training and test datasets contain EEG data from different subjects [7]. The subject-independent approach is more practical because it does not require recalibration for new users. However, subject-independent emotion recognition models often suffer from low performance due to the high variability of EEG signals across individuals. A potential solution to enhance accuracy is incorporating demographics such as cultural information alongside EEG data during model training, as cultural background significantly influences emotional perception and neural responses [8].

By tracking brain activity in response to emotional stimuli, EEG-based emotion recognition models provide valuable insights into how emotions vary across cultures [9]. For instance, Gan et al. [10] explored cultural differences in emotion recognition using EEG and eye movement data, finding that power spectrum density features extracted from the beta frequency band enhanced emotion recognition in Chinese subjects, whereas those extracted from the gamma frequency band improved prediction in French subjects. Liu et al. [11] identified cultural differences in EEG patterns for emotion recognition between Chinese, French, and German subjects, noting that Chinese participants showed reduced activity in the gamma and beta bands, while German and French participants exhibited increased activity in the theta and alpha bands. Similarly, Wu et al. [12] compared EEG-based emotion recognition between Chinese and Germans, finding general similarities in the gamma band but key differences in

delta band activity for Germans during positive emotions.

In general, previous studies suggest that cultural norms and values influence brain activity during emotional processing. However, these studies face some limitations. Many have prioritized improving prediction accuracy, often at the expense of interpretability. Others have focused on developing multimodal approaches by combining EEG and eye movement data, which may obscure the identification of neural-only emotion patterns associated with each cultural background. As a result, the extent to which cultural factors impact EEG-based emotion recognition models remains underexplored. To address this gap, more effort is still needed to develop approaches that rely only on EEG signals to identify emotional neural patterns associated with cultural differences, as well as to measure the impact of cultural background on these predictions.

This study aims to address this issue by initially training deep learning models to detect three emotions across datasets for three different nationalities: French, German, and Chinese. Subsequently, a logistic regression model was employed to correlate the output of the deep learning model with the culture of the subjects, thus analyzing how the cultural factors affect the odds of accurately predicting emotions. The main contributions of this paper to EEG-based emotion recognition are summarized as follows:

- Introducing a predictive approach that enables both emotion recognition and the analysis of culture-based influences on those predictions, utilizing three independent datasets: SEED [13], SEED-FRA [11], and SEED-GER [11].
- Using attention mechanism layers to identify culture-based emotional patterns for emotion prediction across the three independent datasets.
- Using logistic regression to quantify the impact of nationality on the odds for accurately predicting emotions.

## II. METHODS AND MATERIALS

### A. Datasets

In this study, three datasets were utilized: SEED-FRA [11], SEED-GER [11], and SEED [13]. Data collection was performed using a 62-channel ESI NeuroScan system, positioned according to the 10/20 EEG system. These signals were captured at a sampling rate of 1000 Hz and subsequently down-sampled to 200 Hz, ensuring a Nyquist frequency of 100 Hz. Table I shows detail of the datasets, showing the total number of video clip for each emotion.

TABLE I: Detail of the SEED, SEED-GER, and SEED-FRA datasets.

| Dataset | Emotions (Videos) | | | | Subjects | |
|---------|----------|----------|---------|-------|-------------|-------|
| | Positive | Negative | Neutral | Total | Nationality | Total |
| SEED | 15 | 15 | 15 | 45 | Chinese | 15 |
| SEED-GER | 18 | 18 | 18 | 54 | German | 8 |
| SEED-FRA | 21 | 21 | 21 | 63 | French | 8 |

### B. EEG Processing

As EEG signals are prone to noise and artifacts from non-neural sources, we filtered the EEG signals using a band-pass Butterworth filter (0.5-50 Hz). The rationale for using a Butterworth filter was that it can eliminate unwanted frequency components with minimal signal distortion [14]. The selected frequency band for the band-pass filter allowed to maintain the frequencies associated with the five primary brainwave categories, namely delta, $\delta$ (0.5-4 Hz), theta, $\theta$ (4-8 Hz), alpha, $\alpha$ (8-14 Hz), beta, $\beta$ (14-31 Hz), and gamma, $\gamma$ (31-50 Hz).

### C. Feature Extraction

The filtered signals were then segmented into 4-second non-overlapping windows, resulting in a frequency resolution of $0.25\ Hz$. This resolution allowed each segment to capture at least two full cycles for all the frequency bands. Differential entropy (DE) features were extracted from each 4-second segment. To compute DE, it was assumed that the EEG signals followed a Gaussian distribution $x \sim \mathcal{N}(\mu, \sigma^2)$. As such, the DE was computed as:

$$h_b(X) = \frac{1}{2}\ln\left(2\pi e \sigma_b^2\right) \tag{1}$$

where $\sigma_b^2$ is the variance (or power) of the 4-second segment at the $b$-th frequency band, for each 4-second window. DE was calculated for all 62 EEG channels and 5 frequency bands, resulting in a feature vector of size $5 \times 62 = 310$ for each window.

### D. Emotion Recognition Model

Figure 1 shows the deep learning model used to process the extracted feature tensors. The input of this model had dimensions $(B, w, 62, 5)$, where $B$ was the batch size, $w$ was the number of windows, 62 was the number of EEG channels, and 5 was the number of frequencies. The model comprised different modules to process and capture spectral, spatial, and temporal information for emotion recognition.

*1) Spectral processing:* The first module aimed to enhance the spectral information contained in the input features. To spatially correlate the frequency across all the 62 EEG channels, reshaped the input $X$ into a three-dimensional tensor of shape $(B \times w, 62, 5)$ by stacking the four-second segments along the batch dimension. Then, two self-attention mechanism layers were used to refine the spectral information. The first self-attention layer aimed to enhance the frequency values by considering the correlation between EEG channels as follows:

$$M = W_{channels}X, \tag{2}$$

where $W_{channels}$ was computed as $softmax\left(\frac{XX^T}{\sqrt{62}}\right)$. $W_{channels}$ was a 62 squared matrix, where the cell of the intersection between the $i$-th row and the $j$-th column was the correlation scores based on their DE values between each of the $i$-th and $j$-th EEG channels. By multiplying $W_{channels}$ with $X$, the resulting matrix, $M$, of dimension $62 \times 5$ was an enhanced feature matrix where DE features of correlated channels were fused.

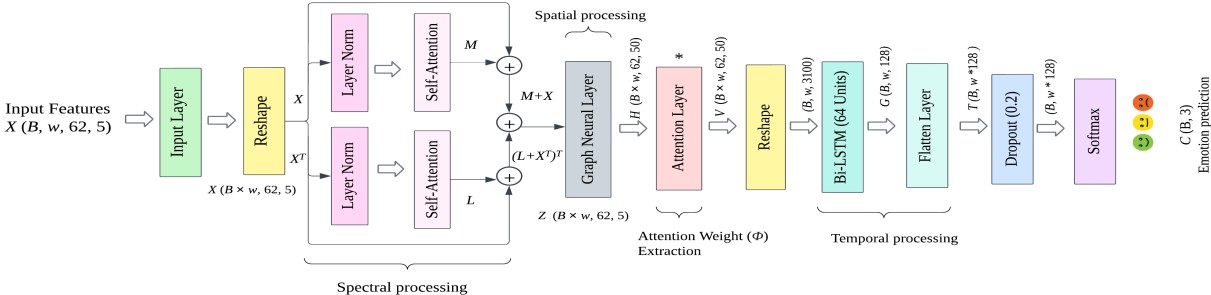

Fig. 1: Diagram of the emotion recognition model. The input tensors have dimensions of $(B, w, 62, 5)$, and $C$ denotes the number of emotion classes: 3 for all the datasets. The $*$ marked attention layer was used for attention weight extraction.

The second self-attention layer was applied to the transposed spectral data, $X^T$, to improve the information of each EEG channel based on the correlation between frequencies. It was computed using the following equation:

$$L = W_{frequency}X^T, \tag{3}$$

The frequency correlation matrix $W_{\text{frequency}}$ was derived as $softmax\left(\frac{X^TX}{\sqrt{5}}\right)$, which captured pairwise relationships among frequency components. The resulting matrix $L$ represented an enriched feature space, where within EEG channels, the DE features were combined based on the similarities among the frequency bands.

To ensure stability during the training process, both the $M$ and $L$ matrices were passed through the normalization and residual connection layers. Finally, to ensure the integration of improved spectral features across EEG channels, the output of the two self-attention layers was fused by using an adding layer as:

$$Z = \left((M + X) + (L + X^T)^T\right). \tag{4}$$

*2) Spatial processing:* For spatial processing, we used a Graph Neural Layer (GNL). The GNL aimed to capture spatial relationships among the refined spectral features in $Z$. The GNL correlated the features as follows:

$$H = \text{LEAKY\_RELU}\left(SZW_{\text{graph}}\right), \tag{5}$$

where $S$ was the adjacency matrix defined as: $S = \tilde{D}^{-\frac{1}{2}}\tilde{A}\tilde{D}^{-\frac{1}{2}}$. $\tilde{A}$ was $A + I$ and $\tilde{D}$ was the degree matrix, with diagonal entries $\tilde{D}_{i,i} = \sum_j \tilde{A}_{ij}$. $I$ corresponded to the identity matrix and $A$ was a 62-square matrix, where each row and column of $A$ corresponded to an EEG channel. The diagonal entries $A_{i,i}$ were set to 0, and the off-diagonal entries $A_{i,j}$ corresponded to the inverse of the Euclidean distance between the $i$-th and the $j$-th EEG channels. The weight matrix $W_{graph}$ was implemented as a dense layer with 50 units and the Leaky Rectified Linear Unit (Leaky ReLU) as the activation function.

*3) Attention Layer:* An attention mechanism layer was used to find the importance of each EEG channel, based on its spatial-spectral features extracted by the GNL. Thus, the output of the GNL, $H$, was passed through the attention layer $G_b(\cdot, \omega_b)$. Assuming that $h_{k,t}$ denotes the feature vector of

the $k$-th EEG channel at the $t$-th tensor, the attention layer $G_b(\cdot, \omega_b)$ mapped $h_{k,t}$ into a hyperbolic space, resulting in $u_{k,t}$. Then, $u_{k,t}$ was passed through a softmax activation function to compute the normalized importance weight for each EEG channel, represented as $\phi_{k,t}$. These weights were subsequently used to calculate the context vector of the $k$-th EEG channel, $v_{k,t}$, at the $t$-th tensor.

$$
\begin{aligned}
u_{k,t} &= \tanh(W_b h_{k,t} + c_b); \\
\phi_{k,t} &= \frac{\exp\left(u_{k,t}^T \cdot u_b\right)}{\sum_k \exp\left(u_{k,t}^T \cdot u_b\right)}; \\
v_{k,t} &= \phi_{k,t} h_{k,t}.
\end{aligned}
\tag{6}
$$

A tensor $V$ with dimensions $(B \times w, 62, 50)$ was formed by arranging the vectors $v_{k,t}$.

*4) Temporal processing:* To capture temporal variations of the features contained in $V$, we reshaped the structure from $(B \times w, 62, 50)$ to $(B, w, 3100)$. This restructuring enabled the features to be placed in a temporally increasing sequence along the second dimension. The resulting structure was fed into a Bidirectional Long Short-Term Memory (BI-LSTM) of 64 units. As a result, the output denoted as $G$, had dimensions $(B, w, 128)$, with 128 representing the concatenation of 64 units from the forward LSTM and 64 units from the backward LSTM. This output captured temporal information from both past and future contexts, making it highly valuable for subsequent prediction tasks. Finally, we used a flattening layer to aggregate the BI-LSTM output, resulting in a 2D structure, $T$, of dimensions $(B, w \times 128)$.

*5) Emotion prediction:* Before making the final emotion predictions, $T$ was fed into a dropout layer with a rate of 0.2 to minimize overfitting. After that, a softmax layer was used to compute class membership probabilities. The emotion class with the highest probability was selected as the predicted class.

*E. Model training and implementation*

The model was evaluated using leave-one-out cross-validation (LOOCV) to ensure a subject-independent approach. During each iteration, model training was done on all subjects except one. This process was repeated for each

subject, resulting in an individual performance metric for each subject.

For each iteration of the LOOCV, we trained the deep learning models by selecting the optimal learning rate through cross-validation on the training set using 20 epochs, testing values from the range $\{1 \times 10^{-4}, 1 \times 10^{-3}, 1 \times 10^{-2}\}$. The best hyperparameters were chosen to retrain a final model on the training set while using cross-entropy as the loss function and applied mini-batch stochastic gradient descent (SGD) with cosine decay for adjusting the learning rate over time [15]. The batch size ($B$) was fixed at 64, and training was carried out for 100 epochs.

### F. Emotion recognition performance

Performance for each emotion class was ratio of correctly predicted samples by total number of samples, resulting in overall accuracy as average accuracy across all three emotions.

*1) Baseline Model:* The baseline model, illustrated in Figure 1, consists of a graph neural network combined with a bidirectional LSTM, designed to directly predict emotions from the input features.

*2) Extended Model:* To evaluate the influence of cultural backgrounds on emotion prediction, the culture variables ($C_1$ and $C_2$) were integrated with the emotion class probabilities generated by the deep learning model. This combined feature set was then passed through a logistic regression model. The cultural variable, representing factors such as country or nationality, was encoded as a binary variable, where '00' denoted Chinese, '01' denoted French, and '10' denoted German.

### G. Impact of subject's culture on emotion predictions

To evaluate the influence of subjects' cultural backgrounds on emotion prediction, the coefficients of the logistic regressor were assessed to evaluate the influence of the predictor variables into the outcome variable. A positive coefficient, $\rho$, indicated an increased likelihood of the desired outcome by a factor of $\exp(\rho)$, while a negative coefficient, $-\rho$, signifies a reduced likelihood by a factor of $\exp(-\rho)$. In our case, the multinomial logistic regression produced a coefficient, $\rho$, which we used to determine whether culture (e.g., Chinese) increased the chances of predicting an emotion class correctly.

Since Leave-One-Out Cross-Validation (LOOCV) was employed to evaluate the model's performance, a separate multinomial logistic regression model was trained for each subject. Upon completing the LOOCV process, the overall (global) coefficient was derived by averaging the individual coefficients across all subjects. The standard error (SE) and the 95% confidence interval were then calculated for each coefficient. Statistical significance was assessed using a two-sided t-test, with the degrees of freedom set to the number of subjects minus one (14 for SEED and 7 for SEED-GER and SEED-FRA).

### H. Attention weight extraction and group-level statistical analysis

In this study, attention weights were extracted to analyze the contribution of different EEG channels in subject-independent emotion recognition (see Figure 1). By extracting and visualizing these attention weights, we generated topographic maps (topo maps) to illustrate the spatial distribution of neural activity. These topo maps provide insights into the brain regions most relevant to emotion classification, aiding in the interpretability of the model's decision-making process.

To assess potential differences in attention weights between Chinese, French, and German, we performed a series of pairwise statistical comparisons. Specifically, for each emotion category, we compared the attention weights between nationality pairs using a two-sample Wilcoxon rank-sum test. Given the multiple comparisons across emotions, EEG channels, and nationality groups, we used Benjamini-Hochberg [16] to reduce the risk of false positives (false discovery rate was set to 0.05).

### I. Ablation Study

We conducted six ablation experiments to evaluate the impact of each component on emotion prediction. First, we tested the model by removing the spectral processing component at the EEG channel level (top branch in Figure 1). Next, we conducted another experiment by removing spectral processing at the frequency band level (bottom branch in Figure 1). In the third and fourth experiments, we excluded all spectral processing and spatial processing (GNN) components respectively, to observe the effects on model performance. Next, we excluded the attention layer to assess its effect on the overall performance. Finally, the temporal processing was removed to assess its impact. In each case, we measured the accuracy and standard deviation across three datasets.

## III. Results

### A. Comparison between baseline and extended model

The results of the accuracy comparison for neutral, positive, and negative emotions across three nationalities: Chinese, German, and French are shown in Figure 2. Both models obtained higher median accuracy rates than the expected accuracy by chance for three classes ($1/3 = 0.33$).

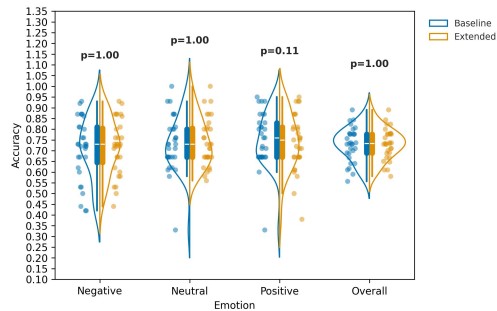

Fig. 2: Distribution of accuracy rates for predicting each emotion class for baseline and extended models; p-values from paired Wilcoxon test.

The violin plots indicate that the baseline and extended models have similar accuracy distributions for neutral and

negative emotions. However, the baseline model outperformed the extended model in predicting positive emotion, although the difference was not statistically significant (Wilcoxon test, p-value = 0.11).

Table II compares our model with other studies that also predicted emotions in the SEED, SEED-FRA and SEED-GER datasets. Our proposed model achieved accuracy rates comparable to those of previous studies, demonstrating its effectiveness and robustness in emotion recognition. For SEED dataset subjects, it achieved 77.2%, surpassing KNN (54.09%), SVM (72.63%), and LR (68.36%) from Liu et al. [11], although slightly behind DNN (82.81%). For SEED-FRA subjects, our model achieved 73.0%, outperforming all baselines, including DNN (64.18%) [11]. For SEED-GER participants, it reached 65.6%, comparable to DNN (65.87%) [11] and surpassing other models. Additionally, our approach exhibited the lowest standard deviation, indicating greater stability in emotion prediction across multiple subjects.

TABLE II: Comparison of emotion classification performance across nationalities using EEG and different models; performance as mean accuracy (Acc.) and standard deviation (SD).

| Nationality | Metric | KNN [11] | SVM [11] | LR [11] | DNN [11] | Ours |
|---|---|---|---|---|---|---|
| Chinese | Acc. | 54.09 | 72.63 | 68.36 | 82.81 | 77.2 |
|  | SD | 8.71 | 10.50 | 11.71 | 7.52 | 5.3 |
| French | Acc. | 37.21 | 50.10 | 47.18 | 64.18 | 73.0 |
|  | SD | 6.76 | 10.29 | 12.20 | 8.56 | 5.0 |
| German | Acc. | 40.94 | 55.64 | 50.39 | 65.87 | 65.6 |
|  | SD | 7.42 | 12.17 | 10.88 | 10.05 | 6.0 |

### B. Impact of subject's culture on emotion predictions

Table III shows the logistic regression coefficients of each variable for the extended model. The odds of accurately predicting negative emotions were significantly higher for German and French subjects compared to Chinese counterparts ($\exp(0.9) \approx 2.46$). This suggests that the probability that the deep learning model correctly identified negative emotions in German and French subjects was approximately 2.46 times greater than in Chinese subjects.

For neutral emotions, the odds of correctly predicting neutral emotions for German subjects was 8.16 times more than Chinese participants ($\exp(2.1) \approx 8.16$, $p$-value $< 0.001$). This indicates a strong distinction in neutral emotion processing between Chinese and German subjects.

For positive emotions, the odds of correctly predicting positive emotions for Chinese subjects were 45% higher compared to French subjects ($\exp(-0.6) \approx 0.55$, $p$-value $< 0.001$). However, no significant differences were observed between German and Chinese participants ($p$-value = 0.489).

### C. Comparing attention weights between Chinese, French, and German

Figure 3 shows the attention weights extracted from Chinese, French, and German subjects, respectively. A more pronounced red intensity indicates a higher attention weight for the corresponding EEG channel and emotion. Moreover, Table IV compares the EEG channels in which attention

TABLE III: Logistic regression coefficients were derived by regressing emotion class on deep model probabilities and a binary-encoded "culture" feature, represented as variables C1 and C2 (Chinese, C1=0 & C2=0; French, C1=0 & C2=1; and German, C1=1 & C2=0).

| Emotion | Ind | Prob. Neutral | Prob. Positive | C1 | C2 |
|---|---|---|---|---|---|
| Negative | Coefficients | -5.3 | -5.2 | 0.9 | 0.9 |
|  | SE | 0.2 | 0.1 | 0.5 | 1.3 |
|  | LCI | -5.4 | -5.2 | 0.7 | 0.4 |
|  | UCI | -5.2 | -5.2 | 1.1 | 1.4 |
|  | p-val | $< 0.001^*$ | $< 0.001^*$ | $< 0.001^*$ | $< 0.001^*$ |
|  | Power | 100% | 100% | 100% | 97% |
| Neutral | Coefficients | 5.4 | 0.1 | 2.1 | 0.3 |
|  | SE | 0.8 | 0.3 | 1.2 | 1.2 |
|  | LCI | 5.1 | 0.0 | 1.7 | -0.1 |
|  | UCI | 5.7 | 0.2 | 2.6 | 0.7 |
|  | p-val | $< 0.001^*$ | 0.034 | $< 0.001^*$ | 0.172 |
|  | Power | 100% | 57% | 100% | 26% |
| Positive | Coefficients | 0.1 | 5.1 | -0.1 | -0.6 |
|  | SE | 0.3 | 0.3 | 0.6 | 0.6 |
|  | LCI | 0.0 | 5.0 | -0.3 | -0.8 |
|  | UCI | 0.2 | 5.2 | 0.1 | -0.3 |
|  | p-val | 0.112 | $< 0.001^*$ | 0.489 | $< 0.001^*$ |
|  | Power | 34% | 100% | 10% | 100% |

weights significantly differed for each emotion type across nationalities.

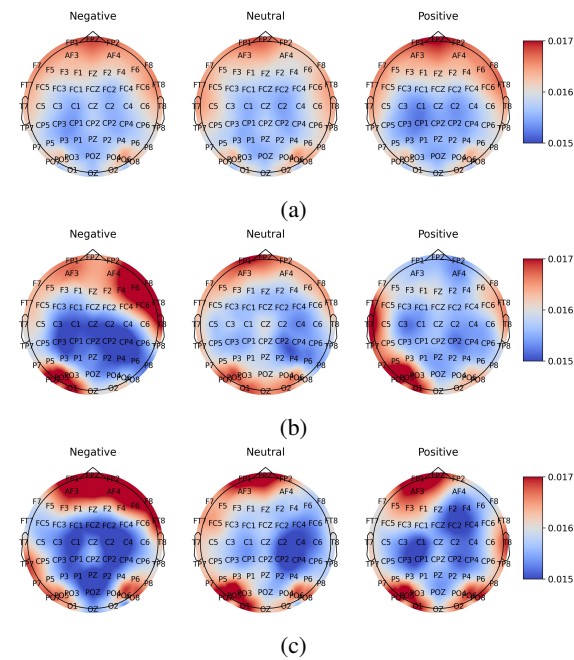

Fig. 3: Topomaps of activation weights given by the Self-Attention Mechanism for each of the 62 EEG nodes for (a) Chinese and (b) French and (c) German subjects.

The activation patterns shown in Figure 3a indicate that Chinese individuals exhibit similar activation for all three emotions. However, the topographic maps for positive emotions appear darker, suggesting stronger activation. Additionally, most activation across all three emotions was concentrated in the frontal region. For negative emotions, a higher concentration of attention weights was observed in the front-right region of the brain, whereas for positive emotions, the weights were primarily located in the left occipital area. As illustrated

in Figures 3b and 3c, activation patterns among French and German participants demonstrated a high degree of similarity. Across both groups, emotional processing exhibited lateralized activations, for negative emotions associated with greater activations in the right frontal region, and positive emotions predominantly involving the left cerebral hemisphere.

The results in Table IV suggest that French and German participants exhibit highly comparable neural activity across all emotion types, with only a minor distinction observed in the processing of neutral emotions at the EEG channel C2. In contrast, more pronounced differences emerged when comparing Chinese participants with the European groups. For negative emotions, significant differences were noted between Chinese and French participants in channels P8 and CP4, and between Chinese and German participants in channel FCz. For neutral emotions, significant differences were observed between Chinese and French participants in the P8, CB1, and C6 channels, and between Chinese and German participants in the FC4 and PO3 channels. For positive emotions, significant differences in attention weights were found in the P5, PO3, O1, and FP2 channels between Chinese and French participants, and in the F8 channel between Chinese and German participants. These findings highlight the presence of distinct neural patterns in Chinese participants when compared to their French and German counterparts.

TABLE IV: EEG channels with significant attention weight differences by emotion and nationality.

| Emotion | Chinese vs French | Chinese vs German | French vs German |
|---|---|---|---|
| Negative | P8, CP4 | FCz | |
| Neutral | P8, CB1, P6, C6 | FC4, C6, PO3 | C2 |
| Positive | P5, PO3, O1, FP2, CB1, PO5, PO7 | F8 | |

*D. Ablation Study*

TABLE V: Ablation study assessing the impact of removing individual components from the deep learning model in Figure 1. Each row presents the mean and standard deviation of LOOCV accuracy, along with comparing to the full model.

| Experiments | SEED | | SEED-GER | | SEED-FRA | |
|---|---|---|---|---|---|---|
| | Acc. (%) | Var. (%) | Acc. (%) | Var. (%) | Acc. (%) | Var. (%) |
| Full Model | 77.62 | – | 64.93 | – | 73.01 | – |
| - Spectral Processing - (EEG channel) | 76.74 | -0.88 | 63.19 | -1.74 | 73.21 | +0.20 |
| - Spectral Processing - (frequency bands) | 74.96 | -2.66 | 58.68 | -6.25 | 71.23 | -1.78 |
| - Spectral Processing | 78.37 | +0.75 | 66.89 | +1.96 | 72.22 | -0.79 |
| - Spatial Processing - GNN | 72.88 | -4.74 | 55.55 | -9.38 | 70.63 | -2.38 |
| - Attention Layer | 65.03 | -12.59 | 60.99 | -3.94 | 66.46 | -6.55 |
| - Temporal Processing | 76.29 | -1.33 | 63.54 | -1.39 | 72.42 | -0.59 |

Table V presents the results of the ablation study, showing the impact of removing different components from the model. Removing spectral processing at the frequency band level led to a significant accuracy drop (-2.66% SEED, -6.25% SEED-GER, -1.78% SEED-FRA). Removing it entirely improved SEED (+0.75%) and SEED-GER (+1.96%) but slightly decreased SEED-FRA (-0.79%). The exclusion of GNN resulted in the largest accuracy loss (-4.74% SEED, -9.38% SEED-GER, -2.38% SEED-FRA), highlighting its importance in capturing spatial dependencies. Removing the attention layer significantly reduced performance, especially in SEED (-12.59%) and SEED-FRA (-6.55%), emphasizing its role in fea-

ture extraction. Eliminating temporal processing led to minor performance drops across all datasets (-1.33% SEED, -1.39% SEED-GER, -0.59% SEED-FRA), indicating a less critical but still relevant contribution. Overall, spatial processing and attention mechanisms have the most substantial impact, while spectral and temporal processing influence performance to varying degrees across datasets.

## IV. DISCUSSION

*A. Main Findings*

Our results highlight the influence of cultural factors on emotion recognition based on EEG signals, with German and French participants showing higher odds of accurately predicting negative emotions compared to Chinese participants. In contrast, Chinese participants showed higher odds of predicting positive emotions relative to their German and French counterparts. These findings suggest potential cultural differences in the neural representation and expression of emotions.

Our findings emphasize that culture plays a role in emotional processing, influencing how emotions are perceived, expressed, and classified based on EEG data. These results suggest that Chinese subjects may be more likely to express positive emotions in comparison to European subjects. Whereas, Germans and French participants may outwardly express negative emotions more than the Chinese. This aligns with research suggesting that individuals from interdependent cultures, such as China, often suppress negative emotions [17]. Understanding these variations can be useful for developing culturally adaptive emotion recognition systems. However, we acknowledge that the logistic regression used in the odds ratio analysis was conducted across overlapping LOOCV folds. As such, the observed differences across nationalities should be interpreted as exploratory rather than definitive statistical inferences.

The topomaps findings suggest distinct neural activation patterns across different cultural groups to predict emotions. Chinese individuals exhibited consistent activation across all three emotions, with stronger responses for positive emotions and predominant frontal region activation. In contrast, French and German individuals showed similar attention weights distributed around the head circumference. Notably, negative emotions were associated with increased attention in the front-right brain region, while positive emotions elicited greater activation in the left occipital area. These findings highlight potential cultural and neurophysiological differences in emotional processing and also suggest brain lateralization may be more pronounced in European populations compared to those of Chinese descent. Brain lateralization means that certain cognitive and emotional processes might be more strongly dominated by one hemisphere of the brain [18]. Thus, for French and German subjects, it is recommended to leverage brain lateralization theory to enhance emotion recognition.

The identified emotional patterns and the influence of nationality on emotion recognition provide valuable insights into how various cultures process emotions. These findings

can help healthcare providers deliver more personalized care, which may improve treatment outcomes for conditions such as depression, anxiety, and autism [19].

### B. Comparison with Previous Studies

Consistent with previous studies on neural patterns across cultures [10]–[12], we found that most attention weight differences were concentrated in the frontal, temporal, and occipital regions. In [10], French participants showed more asymmetric brain connectivity, whereas Chinese participants exhibited a more symmetrical distribution. This observation supports our finding that Chinese individuals display similar neural activation across all three emotions, suggesting a more balanced and consistent pattern of emotional processing. Furthermore, our findings also reveal that French and German individuals exhibited similar attention weight distributions around the head circumference. Aligning with prior research [11], which reported that French and German participants share similar topographical patterns and attentional weight distributions.

Our study also observed differences in emotional processing across nationalities, particularly in the activation patterns of specific EEG channels. This aligns with previous findings, such as those by Wu et al. [12], who reported cultural variations in delta band activity. These results suggest that cultural background influences not only the frequency bands but also the spatial distribution of neural activity, highlighting the importance of considering channel-level differences in cross-cultural emotion recognition.

### C. Limitations and Future Work

We note that the SEED dataset contained more subjects than SEED-GER and SEED-FRA. Thus, as the deep learning model was more exposed to Chinese subjects' data, it could have captured more patterns associated with Chinese subjects. However, the cultural differences in emotional processing highlight and suggest that future research should replicate these results using more balanced nationality ratio datasets.

We also note that the current study focused on emotion datasets with discrete emotion categories and not on the arousal-valence model. To better understand neural patterns of emotional dimensions, future work should validate findings using arousal-valence dataset, such as DEAP [20]. This would allow for an examination of the influence of nationality on the components of arousal and valence. Finally, we acknowledge that only DE features were used in this study; future work should explore alternative or complementary feature representations to enhance model performance.

### V. Conclusion

This study presents a stacking approach combining deep learning and logistic regression to examine the impact of culture on emotion prediction. The results show that culture significantly influences emotional outcomes, showing an effect on emotion processing. European subjects have higher odds of predicting negative emotions, while for Chinese, happy emotions are more likely to be predicted. The cultural effect is similar for both French and German individuals, indicating more shared cultural traits between these groups.

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
