# OpenReview forum: "Evaluating Cultural Impact on Subject-Independent EEG-Based Emotion Recognition Across French, German, and Chinese Datasets"
_IEEE.org/EMBS/BHI/2025/Conference — BHI 2025_

### Official Review · Reviewer_Lr3g · 2025-07-16
**Review of Evaluating Cultural Impact on Subject-Independent EEG-Based Emotion Recognition Across French, German, and Chinese Datasets**

**Confidence:** 3
**Clarity Of Writing:** great
**Clinical Significance:** fair
**Methodological Novelty:** fair
**Overall Rating:** 6
**Final Rating:** 7

**Experiments And Results:**

good

**Questions For The Authors:**

Please include a reasoning for decomposing the EEG into DE features, did you compare with more diverse feature sets or with raw EEG as input?
What value was used for w, the number of windows input to the model? Does it depend on the length of each recording?
In the caption of Figure 1 C is referenced, I can't understand where C is shown in this figure? Can this be addressed this or highlight the location of C in the figure?
Do you think that facial expressions (muscle artifacts) impact the activations of electrodes in the frontal regions (see figure 3)?

**Strengths:**

This is an interesting work, and the paper is well structured. A range of different analysis experiments are included to try to deepen understanding of the cultural differences between the studied cohorts and to highlight the different features of the architecture that impact performance.

**Summary Of The Paper:**

This paper presents the differences in performance and in the learned internal representations of models trained on EEG emotion datasets for different participants from different cultural backgrounds. A deep learning architecture is designed, which utilizes self-attention, graph layers, and LSTM layers to process the contextual, spatial, and temporal information in the EEG signals. The EEG signals are transformed into features sets represented by their differential entropy in each frequency band (alpha, beta, gamma etc.).
They extend their base model by training a logistic regression model that takes the output of the base model plus the cultural background of a participant. The impact of a participant's culture is assessed by examining the learned components of the logistic regression model. Additionally an ablation study was used to show which parts of the base deep learning architecture contribute to performance accuracy.
The performance reported in this study appears to be an improvement or equivalent to the performances reported in literature, but the standard deviation of performance accuracy is lower for the architecture proposed here.

**Weaknesses:**

The base architecture utilized in this work has a lot of different components, is this based on previous works? Reference [11] reports the performance of a DNN, is this architecture similar to that used in this work. If it is completely novel then this could be highlighted.

It is unclear from this work whether there was a positive impact in the extended model (including the logistic regression), could the results in table II indicate whether "Ours" is the baseline or extended model. Or alternatively both performances could be shown. This would allow the authors to further discuss the impact of the extended model; this could support the plots in figure 2 which appear to show very little difference.

---

### Official Review · Reviewer_vRkf · 2025-07-17
**Exploring cultural impact on emotion recognition is important, but evaluation lacks robustness**

**Confidence:** 5
**Clarity Of Writing:** fair
**Clinical Significance:** great
**Methodological Novelty:** fair
**Overall Rating:** 5
**Final Rating:** 6

**Experiments And Results:**

fair

**Questions For The Authors:**

1-Could the authors provide the class-wise label distributions or F1-scores to justify the use of average accuracy across emotion classes?

2- How do the authors justify the use of t-tests on logistic regression coefficients trained on overlapping LOOCV folds, given the potential dependence between samples?

3- Why did the authors opt for a two-stage model where softmax outputs are combined with binary cultural variables in a logistic regression? Was any normalization or calibration applied to ensure compatibility between these heterogeneous features?

4- How does the proposed method mitigate overfitting risk, particularly given the low sample sizes in SEED-FRA and SEED-GER?

**Strengths:**

1.	The paper tackles an underexplored yet important issue by analyzing how nationality influences EEG-based emotion classification, offering valuable insight into the role of cultural background in affective computing.

2.	The inclusion of three different nationalities (Chinese, French, and German) across datasets allows the authors to make cross-cultural comparisons, which is rarely done in EEG emotion recognition studies.


3.	The experimental setup using LOOCV ensures that the evaluation remains subject-independent, which is a key challenge in EEG-based affective modeling

**Summary Of The Paper:**

This paper introduces a novel deep-learning architecture for subject-independent EEG-based emotion recognition that integrates demographic features, age, sex, and notably nationality, via an attention-based fusion mechanism. Using leave-one-subject-out cross-validation on three culturally diverse datasets, the model achieves modest gains in overall accuracy, with more pronounced improvements for certain emotions and demographic subgroups.

**Weaknesses:**

1.	Averaging class-wise accuracy can be used as a secondary metric if classes are reasonably balanced, but without confirming class balance, it is not sufficient alone. Authors should report F1-score (macro and per-class) or at least provide the class distributions to support the use of average accuracy across classes.

2.	While the paper attempts to evaluate cultural influence on emotion prediction via logistic regression coefficients, I have concerns about the robustness of this analysis. Specifically, the authors train a separate logistic regressor in each LOOCV fold and then average the coefficients to estimate global cultural effects, performing a t-test to assess statistical significance. However, this approach violates key assumptions of the t-test: the coefficients come from overlapping training data and are therefore not independent. Moreover, the small and imbalanced group sizes, particularly in SEED-FRA and SEED-GER, further reduce statistical power and increase the risk of misleading results. To more reliably estimate the effect of culture, I recommend the use of mixed-effects models or bootstrap-based methods, which are better suited for small and dependent data structures. Without these corrections, the reported significance of cultural influence should be interpreted with caution.


3.	The extended model combines the softmax probabilities output by the deep learning model with binary-encoded cultural variables (C1 and C2), which are then passed into a separate logistic regression model. This architecture breaks end-to-end learning and treats culture as a post hoc correction to the model’s output, rather than integrating it meaningfully into the learning process. The results in Figure 2 further support the concern with this two-stage architecture: adding cultural variables post hoc to the softmax outputs fails to produce any statistically significant gains in accuracy. This highlights a key limitation of treating culture as a correction term rather than integrating it directly into the representation learning process.

4.	The concept of integrating demographics with EEG for emotion recognition is not entirely new, and more principled end-to-end models already exist. For example, Peng et al. (2023) use a graph convolutional network with built-in attention that jointly processes EEG signals and sex labels during training, enabling sex-informed representation learning rather than demographic adjustment post hoc (https://pubmed.ncbi.nlm.nih.gov/37906969/). The domain-adversarial graph attention model (DAGAM) explicitly tackles subject variability using graph structure, self-attention pooling, and adversarial training to achieve state-of-the-art cross-subject emotion recognition (https://arxiv.org/abs/2202.12948). I also recommend reviewing the survey by Khalid & Willis (2022) (https://arxiv.org/abs/2201.06610), which highlights SOTA methods for emotion prediction. To substantiate the claimed contribution here, I recommend the authors implement and benchmark against at least one of these integrated architectures to demonstrate clear improvements in accuracy, interpretability, or fairness.

---

### Official Review · Reviewer_d2BE · 2025-07-18
**The paper explores the impact of cultural background on EEG-based emotion recognition, using subject-independent models across Chinese, French, and German datasets. While the motivation is timely and relevant, the methodological novelty and empirical insights are somewhat limited.**

**Confidence:** 4
**Clarity Of Writing:** good
**Clinical Significance:** fair
**Methodological Novelty:** fair
**Overall Rating:** 6

**Experiments And Results:**

good

**Questions For The Authors:**

Why was cultural information only used in a separate logistic regression instead of being integrated into the deep learning architecture (e.g., via conditional modulation)? Integrating culture into the core model could enhance interpretability and performance.

Were any steps taken to balance the datasets or normalize for class imbalance across nationalities? How might the imbalance have influenced the logistic regression results?

How consistent are the findings when using alternative emotional models (e.g., arousal/valence)? Would love to see how these results hold using datasets like DEAP.

How would the findings change if a cross-cultural model was trained across all groups together vs. training individual models per culture? That might better test generalizability.

How confident are you that differences in activation are due to culture and not low-level factors like video content or electrode montage variation?

**Strengths:**

Addresses an important and underexplored issue: cultural impact on emotion recognition.

Employs a relatively well-structured deep learning model incorporating spectral, spatial (via GNN), and temporal features.

Use of multinomial logistic regression adds interpretability to the impact of cultural variables.

Attention-based topographic mapping adds a layer of neuroscientific interpretability.

Ablation study is comprehensive and reinforces the importance of specific model components.

**Summary Of The Paper:**

This study investigates the influence of culture on EEG-based emotion recognition by applying a stacking approach that integrates deep learning (with attention mechanisms and GNNs) and logistic regression. EEG data from three datasets—SEED (Chinese), SEED-FRA (French), and SEED-GER (German)—were used to predict three discrete emotions (positive, neutral, negative). The deep learning model extracted spectral, spatial, and temporal features, while the logistic regression layer quantified the effect of nationality on prediction odds. Attention maps were used to interpret spatial neural activation, and ablation studies evaluated the contribution of model components.

**Weaknesses:**

The inclusion of cultural information (i.e., nationality) is not deeply integrated into the deep model itself—only used post hoc via logistic regression. This limits methodological novelty.

The model architecture is adapted from common EEG classification paradigms and does not introduce a novel approach for culture-aware modeling.

The cultural effect size is modest and often lacks significance (e.g., positive emotion classification between German and Chinese subjects).

No analysis of arousal/valence dimensions, which are often more informative than discrete emotions in affective neuroscience.

Dataset imbalance (15 Chinese vs. 8 each of French and German) likely biases the results; this is acknowledged but not corrected for.

Interpretation of results often overstates the cultural implications without controlling for confounding factors such as language, gender balance, or media content.

The use of LOOCV for all models may not generalize well for small datasets—confidence intervals should be interpreted cautiously.

---

### Official Review · Reviewer_3dP7 · 2025-07-21
**Review of Evaluating Cultural Impact on Subject-Independent EEG-Based Emotion Recognition Across French, German, and Chinese Datasets**

**Confidence:** 4
**Clarity Of Writing:** good
**Clinical Significance:** fair
**Methodological Novelty:** fair
**Overall Rating:** 7
**Final Rating:** 7

**Experiments And Results:**

fair

**Questions For The Authors:**

1. It’s unclear whether the cultural variables (C1, C2) are included during inference or used only for post hoc analysis via logistic regression. If the extended model incorporates cultural data directly at prediction time, it raises concerns about generalizability and practicality, especially in real-world deployments where such metadata may be unavailable. Clarifying this could significantly affect my assessment of the model's applicability and fairness.

2. The SEED dataset has nearly double the number of subjects compared to SEED-FRA and SEED-GER, possibly skewing the model toward Chinese-specific patterns. Was there attempt to balance this via sampling, weighting, or separate normalization?

3. While the topographic maps are visually informative, the interpretation of attention weights is not deeply linked to known neuroanatomical emotion pathways or prior EEG studies. More contextualization could strengthen claims about neural pattern differences.

4. The use of only three emotion classes (positive, neutral, negative) may limit the model’s broader applicability. Have the authors considered adapting their approach to dimensional emotion models (e.g., arousal-valence, DEAP dataset)?

**Strengths:**

The paper addresses an under explored yet highly impactful research question: how cultural background influences subject-independent EEG-based emotion recognition. This focus is particularly relevant to the development of fair, inclusive, and generalizable emotion AI systems, where cultural bias remains a critical challenge. By explicitly modeling cultural factors, the authors move beyond traditional accuracy-centric approaches and contribute to a more nuanced understanding of affective computing across populations.

A key strength lies in the proposed stacking framework that integrates a deep learning model—incorporating spectral, spatial, temporal, and attention-based mechanisms—with a multinomial logistic regression layer. This combination not only boosts emotion recognition performance but also enhances model interpretability by quantifying the impact of cultural variables on classification outcomes. The use of attention mechanisms further aids in identifying relevant EEG regions, adding physiological insight to the findings.

The model is evaluated across three culturally distinct EEG datasets, enabling robust cross-cultural comparisons. The analysis is comprehensive, including ablation studies, statistical significance testing, and topographic attention visualizations. These elements support both interpretability and reproducibility of results. Overall, the paper demonstrates methodological rigor and makes a meaningful contribution to the literature on ethical and culturally aware emotion recognition.

**Summary Of The Paper:**

This paper presents a novel method to assess the impact of cultural background on EEG-based emotion recognition in a subject-independent setting. The authors introduce a deep learning model incorporating spectral, spatial, temporal, and attention mechanisms to classify emotions (positive, neutral, negative) across three nationalities (Chinese, French, German), using the SEED, SEED-FRA, and SEED-GER datasets.
A stacking approach integrates this model with a multinomial logistic regression to quantify cultural influences on prediction odds. Results show that including cultural information improves recognition performance, especially for Chinese (positive emotions) and European participants (negative emotions). Attention map analysis reveals culturally distinct neural activation patterns. The work demonstrates the value of integrating demographic context in affective computing and contributes to interpretable and inclusive emotion recognition systems.

**Weaknesses:**

One key limitation is the imbalance in dataset sizes, with significantly more Chinese participants than French or German. This may bias the model’s learning and undermine the strength of the cross-cultural conclusions. While acknowledged, the paper lacks mitigation strategies. The emotion model is also limited to three discrete categories, which may oversimplify affective states and reduce ecological validity. Additionally, it is unclear whether cultural variables are included during inference or only in post hoc analysis, making the distinction between baseline and extended models somewhat ambiguous. More detail on reproducibility, such as full architecture parameters, computational resources, and training time is also needed for replication. Finally, while the attention maps provide insight, there is limited discussion relating them to established neurophysiological findings.